# Human Bone Marrow-Derived Mesenchymal Stromal Cells Reduce the Severity of Experimental Necrotizing Enterocolitis in a Concentration-Dependent Manner

**DOI:** 10.3390/cells12050760

**Published:** 2023-02-27

**Authors:** Livia Provitera, Andrea Tomaselli, Genny Raffaeli, Stefania Crippa, Cristina Arribas, Ilaria Amodeo, Silvia Gulden, Giacomo Simeone Amelio, Valeria Cortesi, Francesca Manzoni, Gaia Cervellini, Jacopo Cerasani, Camilla Menis, Nicola Pesenti, Matteo Tripodi, Ludovica Santi, Marco Maggioni, Caterina Lonati, Samanta Oldoni, Francesca Algieri, Felipe Garrido, Maria Ester Bernardo, Fabio Mosca, Giacomo Cavallaro

**Affiliations:** 1Neonatal Intensive Care Unit, Fondazione IRCCS Ca’ Granda Ospedale Maggiore Policlinico, 20122 Milan, Italy; 2Department of Clinical Sciences and Community Health, Università degli Studi di Milano, 20122 Milan, Italy; 3San Raffaele Telethon Institute for Gene Therapy (SR-Tiget), IRCCS San Raffaele Scientific Institute, 20132 Milan, Italy; 4Department of Pediatrics, Clínica Universidad de Navarra, 28027 Madrid, Spain; 5Neonatal Intensive Care Unit, Sant’Anna Hospital, 22042 Como, Italy; 6Department of Statistics and Quantitative Methods, Division of Biostatistics, Epidemiology and Public Health, University of Milano-Bicocca, 20126 Milan, Italy; 7Revelo Datalabs S.R.L., 20142 Milan, Italy; 8Department of Pathology, Fondazione Ca’ Granda Ospedale Maggiore Policlinico, 20122 Milan, Italy; 9Center for Preclinical Investigation, Fondazione IRCCS Ca’ Granda Ospedale Maggiore Policlinico, 20122 Milan, Italy; 10Research and Development Unit, Postbiotica S.R.L., 20123 Milan, Italy; 11Pediatric Immunohematology Unit, BMT Program, IRCCS San Raffaele Scientific Institute, 20132 Milan, Italy; 12Maternal and Child Department, Vita-Salute San Raffaele University, 20132 Milan, Italy

**Keywords:** necrotizing enterocolitis, neonate, inflammation, human bone marrow mesenchymal stromal cells, mouse model, apoptosis, zonula occludens-1, interleukin 1b, caspase 3

## Abstract

Necrotizing enterocolitis (NEC) is a devastating gut disease in preterm neonates. In NEC animal models, mesenchymal stromal cells (MSCs) administration has reduced the incidence and severity of NEC. We developed and characterized a novel mouse model of NEC to evaluate the effect of human bone marrow-derived MSCs (hBM-MSCs) in tissue regeneration and epithelial gut repair. NEC was induced in C57BL/6 mouse pups at postnatal days (PND) 3–6 by (A) gavage feeding term infant formula, (B) hypoxia/hypothermia, and (C) lipopolysaccharide. Intraperitoneal injections of PBS or two hBM-MSCs doses (0.5 × 10^6^ or 1 × 10^6^) were given on PND2. At PND 6, we harvested intestine samples from all groups. The NEC group showed an incidence of NEC of 50% compared with controls (*p* < 0.001). Severity of bowel damage was reduced by hBM-MSCs compared to the PBS-treated NEC group in a concentration-dependent manner, with hBM-MSCs (1 × 10^6^) inducing a NEC incidence reduction of up to 0% (*p* < 0.001). We showed that hBM-MSCs enhanced intestinal cell survival, preserving intestinal barrier integrity and decreasing mucosal inflammation and apoptosis. In conclusion, we established a novel NEC animal model and demonstrated that hBM-MSCs administration reduced the NEC incidence and severity in a concentration-dependent manner, enhancing intestinal barrier integrity.

## 1. Introduction

NEC represents the first gastroenterological urgency in neonatal intensive care units (NICUs), affecting 5–7% of preterm neonates with a mortality rate of 30–40% [1,2,3,4,5]. The disease may present various clinical signs and symptoms, with non-specific manifestations (e.g., apnea, bradycardia, lethargy) and feeding intolerance, often making early diagnosis difficult [6]. Although some NEC cases can be managed medically, 20–60% of infants with NEC require surgery, with a mortality rate of around 35% and potential postoperative complications, such as wound dehiscence, intra-abdominal abscesses, intestinal stenosis, and “short bowel syndrome” [1,7,8]. In addition, survivors face lifelong complications, such as adverse neurodevelopmental outcomes [9].

The mortality rate for NEC has not significantly changed over the years, reaching peaks of 30–50% in Extremely Low Birth Weight (ELBW) infants [10]. Although the condition has been extensively studied and epidemiology strongly suggests a multifactorial etiology, the pathogenesis of NEC remains to be fully identified, making current preventive and therapeutic strategies insufficient [11].

Several animal models have been developed to uncover the pathogenesis of NEC and identify new preventive and therapeutic approaches, considering potential risk factors such as intestinal immaturity, hypoxic–ischemic insults, formula feeding, and bacterial contribution [12,13,14,15,16,17,18,19,20,21].

Enteral formula feeding represents a commonly recognized risk factor for NEC [22,23]. Immaturity of the intestinal barrier, combined with microbial dysbiosis and impaired gastrointestinal motility, contributes to food intolerance, creating a substrate for the occurrence of NEC [24]. In addition, the osmolality of enteral feeds, generally higher than human breast milk (450 mOsm/kg and 300 mOsm/kg, respectively), could modulate the inflammatory response leading to mucosal damage [25]. Most NEC animal models rely on the administration of a combination of canine puppy milk replacer with human formula milk in varying proportions, resulting in a protein-rich milk that differs in composition from human breast milk [26].

In addition to hyperosmolar formula feedings, most small animal models use oral administration of LPS of *Escherichia coli* with repeated hypoxia and/or hypothermia to induce intestinal inflammation [17,27,28,29,30,31,32,33]. NEC-associated dysbiosis can be caused by many factors, including cesarean section, broad-spectrum antibiotics, or infant formula, immaturity of the neonatal immune response, and antiacid drugs [25,34,35,36,37,38].

The intestinal immune system, composed of the intestinal epithelium, immune cells, and commensal bacteria, plays a crucial role in maintaining homeostasis and defending against pathogens [39].

The epithelium consists of villi and crypts with different types of cells, while the lamina propria beneath it houses the immune cells (intraepithelial lymphocytes, regulatory T cells, T cells, B cells, macrophages, and dendritic cells) [40].

Increased Toll-like receptor (TLR) 4 expression in the premature intestine is pivotal in the pathogenesis of NEC and is activated by the LPS of Gram-negative bacteria [41,42,43].

The interaction between TLR4 and LPS breaks down the intestinal barrier and allows the translocation of pathogenic bacteria. This leads to a pro-inflammatory response, with an increased production of pro-inflammatory cytokines (IL-6, IL-8, and TNF) as well as an increase in Th17 cells, and a decrease in Tregs, leading to a profound inflammatory response and subsequently NEC [39].

Despite many advances, currently available preclinical models of NEC are unable to fully reproduce the clinical, macroscopic, and histopathological features of the disease [44,45].

To overcome this limitation, we developed a novel mouse model of NEC based on gavage-feeding with term infant formula, hypoxia, hypothermia, and oral administration of LPS.

Stem cell-based therapies are rapidly emerging as potential treatments for neonatal diseases. Promising preclinical results have been obtained with mesenchymal stromal cells (MSCs) [46,47]. Human MSCs are multipotent cells found in various tissues, including bone marrow (BM), adipose tissue, umbilical cord blood, and placenta, from which they can be easily isolated and expanded ex vivo for clinical use [48,49,50]. Furthermore, MSCs exhibit several exclusive features that make them promising candidates for stem cell-based regenerative therapy. These properties include the ability to sense inflammatory signals via TLRs and secrete several anti-inflammatory molecules to support tissue regeneration, regulate immune responses to protect tissues from excessive damage, and promote the survival of tissue-resident progenitors when injected in response to injury [51,52,53,54,55,56,57,58,59,60,61,62,63]. In addition, MSCs lack the expression of HLA class II and co-stimulatory molecules, making them immune-evasive and allowing the administration of third-party allogeneic MSCs [64]. Based on their functional properties, MSCs may represent an attractive therapeutic candidate for treating inflammatory neonatal diseases, including NEC [64].

In this study, we established a novel mouse model of neonatal NEC and assessed the regenerative role of human BM-MSCs to attenuate the severity of experimental NEC.

## 2. Materials and Methods

### 2.1. Animals

Experiments were performed according to the recommendations of the *Guide for the Care and Use of Laboratory Animals of the National Institutes of Health* at the Center of Preclinical Investigation, Fondazione IRCCS Ca’ Granda Ospedale Maggiore Policlinico, Milan, Italy. The experimental protocol was approved by the Italian Institute of Health (Number: 935/2018-PR).

Adult 14-day pregnant C57BL/6SnJ female mice (Charles River, Lecco, Italy) were housed in a ventilated cage system at 22 ± 2 °C, on a 12 h dark/light cycle, allowed free access to mouse chow feed, and water ad libitum.

Experiments were conducted on 2- to 6-day-old, naturally delivered mouse pups.

### 2.2. Isolation and Expansion of Human BM-MSCs

Human BM-MSCs were isolated from residual BM aspirates of healthy donors (median age = 12 years; range 4–18 years) who donated BM for transplantation at San Raffaele Scientific Institute after obtaining informed consent according to a protocol approved by the San Raffaele Ethical Committee (Tiget09).

MSCs clones were obtained by plastic adherence according to a previously published protocol [65]. hBM-MSCs were expanded as fibroblast-like cells in DMEM + GlutaMAX (Thermo Fisher, Waltham, MA, USA) supplemented with 5% platelet lysate and 1% penicillin/streptomycin.

When reaching 80% of confluence, BM-MSCs were split at a concentration of 2500 cells/cm^2^. BM-MSCs at a cell passage between 3 and 6 were used for transplantation.

Twenty-four hours before NEC induction, we intraperitoneally transplanted two different MSCs doses (0.5 × 10^6^ and 1 × 10^6^) into mouse pups. Considering the high variability associated with MSC donors, we transplanted a pool of 3 different hBM-MSCs.

### 2.3. Human BM-MSC Characterization

Ex vivo expanded BM-MSCs were characterized for expressing MSC official markers by flow cytometry and their ability to differentiate into mesodermal cell types, as indicated by the International Society for Cellular Therapy. In detail, BM-MSCs were trypsinized, washed in phosphate-buffered saline (PBS) + 2% fetal bovine serum (FBS), and stained with the following antibodies for 15 min at room temperature: CD105 FITC, CD73 PE, CD90 PE, CD146 PB, CD45 FITC, CD31 FITC (BioLegend, San Diego, CA, USA). After washing twice in PBS + 2% FBS, cells were analyzed using the BD FACSCanto^TM^ (BD Life Sciences, San Jose, CA, USA). Data were analyzed using FlowJo™ v10.8 Software (BD Life Sciences, Ashland, OR, USA) and represented as a percentage of positive cells on total cells.

BM-MSCs were induced to differentiate into osteoblasts and adipocytes using the proper differentiation medium for human cells (Miltenyi Biotec, Bergisch-Gladbach, Germany). After 14 days, adipogenic and osteogenic differentiation was evaluated by Oil Red O (Sigma-Aldrich, St. Louis, MO, USA) and Alizarin Red staining (Sigma-Aldrich, St. Louis, MO, USA), respectively.

### 2.4. RNA Extraction and RT-PCR

According to the manufacturer’s instructions, RNA extracts from BM-MSCs were obtained using the RNeasy Micro Kit (QIAGEN GmbH, Hilden, Germany). DNase treatment was performed using RNase-free DNase Set (QIAGEN GmbH, Hilden, Germany). Moreover, RNA extracts from mouse intestines were obtained using the PureLink RNA Mini kit (Invitrogen, Waltham, MA, USA). DNase treatment was performed using the TURBO DNA-free Kit (Invitrogen, Waltham, MA, USA). cDNA was synthesized from 1 μg total RNA using the high-capacity reverse transcription kit (Applied Biosystems, Waltham, MA, USA). SYBR Green-based quantitative PCR was performed using QuantiFast SYBR Green PCR Kit (QIAGEN GmbH, Hilden, Germany) starting from 10 ng of cDNA with a Viia7 real-time PCR system (Thermo Fisher, Waltham, MA, USA). In addition, expression levels for zonula occludens-1 (ZO-1), Interleukin 1b (IL1b), and B-cell lymphoma 2 (Bcl2) were calculated by the ΔΔCt and normalized to reference housekeeping gene Actin beta (ACTB). Primer sequences used in this study are listed in Table 1.

### 2.5. Mouse Intestine Immunohistochemistry (IHC)

Intestinal samples were routinely formalin fixed-paraffin embedded according to a standardized protocol and cut into 5 µm-thick slices. Intestine sections were stained with the following antibodies: active mouse caspase-3 (R&D System, Minneapolis, MN, USA) and Ki67 antibody (R&D System, Minneapolis, MN, USA). In addition, quantifying Ki67 positive nuclei was performed on 3 sections/samples using the Image J software (*n* = 5).

### 2.6. Experimental NEC Neonatal Mouse Model

NEC was induced using slight modifications of the protocol used by Tian et al., based on gavage-feeding with term infant formula, hypoxia, hypothermia, and oral administration of LPS [66]. Pups were randomly assigned to the following experimental groups: (a) control (*n* = 16), (b) experimental NEC (*n =* 26), (c) NEC + PBS (*n =* 15), (d) NEC + hBM-MSCs (0.5 × 10^6^) (*n =* 23) and (e) NEC + hBM-MSCs (1 × 10^6^) (*n =* 15). Control pups were left with their mothers, breastfed ad libitum, and were not submitted to any stress or treatment.

On postnatal days (PND) 2, pups belonging to the experimental arms NEC + PBS, NEC + hBM-MSCs (0.5 × 10^6^), and NEC + hBM-MSCs (1 × 10^6^) were intraperitoneally injected with 50 μL of sterile PBS 1x (a vehicle control), hBM-MSCs (0.5 × 10^6^)/50 μL or hBM-MSCs (1 × 10^6^)/50 μL (Figure 1), respectively.

Pups exposed to the experimental NEC protocol were dam-separated on PND 3 (approximately 72 h after birth) and placed in a neonatal incubator (Dräger, Lübeck, Germany) at 33 °C, with 60% of humidity, on a 12-h-dark/light cycle.

Pups from the groups subjected to experimental NEC were gavage fed every 3 h with the standard formula for term babies using a 1.9 F silastic catheter (Vygon, Aachen, Germany) starting 1 h after dam separation. Daily feeding volume increased as tolerated (30, 40, and 50µL on PND 3, 4, and 5, respectively) (Figure 1). Orogastric gavage was performed by introducing the silastic catheter in the esophagus of the pup by direct vision, connected to a 1 mL syringe filled with the exact volume of milk to administer. Before every administration, the catheter was primed due to the small volumes of milk used [66,67,68,69,70].

Mice were subjected to hypoxia/hypothermia twice per day (every 12 h): 60 s in 100% nitrogen (N_2_), followed by 10 min at 4 °C (Figure 1).

In addition, 18 h after dam separation, pups were gavage fed with 3 mg/Kg of *E. coli* 055: B5 LPS (Sigma-Aldrich, St. Louis, MO, USA), dissolved in sterile water for injections (Figure 1) [67,68,69].

Animals were euthanized 72 h after dam separation (PND6) or earlier in case of distress or early clinical signs of NEC. Mice subjected to NEC and injected with PBS were used as a positive control. Breastfed mice, not subjected to NEC, were used as a negative control.

### 2.7. Bodyweight and Survival Rate

Bodyweight was assessed daily, starting on PND 2. In addition, the number and the approximate time of deaths were recorded daily for each experimental group.

### 2.8. Histological Injury Score of NEC in Neonatal Mice

The intestine was harvested, fixed in 10% buffered formalin, embedded in paraffin, sectioned (4 µm), and stained with hematoxylin–eosin using an internal protocol. In addition, small and large intestinal tissues were investigated histologically for intestinal damage by a histopathologist blinded to the experimental conditions.

The severity of the experimental NEC was classified from 0 to 4 using a previously validated scoring system [66].

The definition for each histological grade is as follows: grade 0: intact villi; grade 1: superficial epithelial cell sloughing; grade 2: high number of karyorrhectic nuclei, partially compromised villi structure, basal cells still present; grade 3: complete villous necrosis, but basal cells still appreciable; grade 4: transmural necrosis, basal membrane destroyed.

Samples with histological scores of 2 or higher were considered positive for NEC. Severe NEC was defined as an injury score of grade 3 or 4 [71].

### 2.9. Statistical Analysis

Bodyweight changes were presented as means ± SEM, and two-way ANOVA was used to study differences in growth between experimental groups over time. Survival data were represented by Kaplan–Meier curves and compared by log-rank (Mantel–Cox) test. NEC incidence was compared using Fisher’s Exact test and expressed as a percentage. Gene expression analysis and evaluation of Ki67 positive nuclei are represented as means ± SEM. NEC mice treated with hBM-MSCs were compared to NEC mice injected with PBS as controls. Mann–Whitney test was used for statistical evaluation (*** *p* ≤ 0.001; ** *p* ≤ 0.01; * *p* ≤ 0.05). *p*-values less than 0.05 were considered significant. Graphs and *p*-values were obtained with GraphPad Prism 9 (GraphPad, La Jolla, CA, USA).

## 3. Results

### 3.1. The Present NEC Model Based on Term Infant Formula Feedings Can Induce the Pathological Hallmarks of the Disease

In this study, we set up a new neonatal mouse model of experimental NEC based on previously published methods with slight modifications [43,66]. Representative histologic NEC injury grades are shown in Figure 2.

#### 3.1.1. Survival Rate

The first analyzed endpoint was survival at 72 h. The survival rates observed for the two experimental groups were as follows: control, 100% (16/16); NEC, 38.5% (10/26).

Mouse pups who underwent the NEC protocol (NEC group) showed a significantly lower survival rate compared with breastfed pups (*p* ≤ 0.0001; Log-rank Mantel–Cox test) Appendix A.

#### 3.1.2. Body Weight

The starting body weight of mouse pups was comparable across experimental groups before the study on PND 2 and PND 3 (control vs. NEC, not statistically significant). However, control mice showed a remarkable increase in body weight from PND 2 to PND 6. The body weights in the NEC group were dramatically lower (day 4, *p* < 0.0001; day 5, *p* < 0.0001; day 6, *p* < 0.0001; two-way ANOVA with Sidak multiple comparisons post-hoc test) than those in the control group (Table 2).

#### 3.1.3. NEC Incidence and Histological Score

The disease developed in 50% (13/26) of pups subjected to experimental NEC (Figure 3A). In particular, 23.1% (*n =* 6) of pups showed a grade 2 NEC, while 26.9% (*n =* 7) showed a grade 3 NEC (Figure 3A). No pups (0/16) in the control group developed any signs of NEC (*p* = 0.0005, Fisher’s Exact test) (Figure 3A).

Controls showed normal development and healthy intestine compared with pups that underwent experimental NEC (Figure 3B).

### 3.2. Treatment with hBM-MSCs Decreases the Incidence and Severity of Experimental NEC in a Concentration-Dependent Fashion in Neonatal Mice

#### 3.2.1. Survival Rates

All unstressed, control pups survived the 72-h protocol (Appendix A). Pups injected with hBM-MSCs (1 × 10^6^) displayed no significant differences in survival compared to control pups, showing a survival rate of 100% (15/15) at 72 h (Appendix A). On the other hand, pups undergoing NEC protocol and injected with PBS alone or with hBM-MSCs (0.5 × 10^6^) exhibited a decreased survival compared to the breastfed group, with a survival rate of 20% (3/15) and 47.8% (11/23), respectively, (Appendix A) (Control vs. NEC + PBS, *p* < 0.0001; Control vs. NEC + hBM-MSCs (0.5 × 10^6^), *p* = 0.0008; NEC + PBS vs. NEC + hBM-MSCs (1 × 10^6^), *p* < 0.0001; NEC + hBM-MSCs (0.5 × 10^6^) vs. NEC + hBM-MSCs (1 × 10^6^), *p* = 0.0004; Log-rank Mantel–Cox test).

#### 3.2.2. Body Weight

All the pups belonging to the four groups exposed to experimental NEC showed decreased body weight starting from PND 5 compared to the control group (Table 2) (PND5-PND6: NEC + PBS *p* < 0.0001; NEC + hBM-MSCs (0.5 × 10^6^) *p* < 0.0001; NEC + hBM-MSCs (1 × 10^6^) *p* < 0.0001 (two-way ANOVA, with Sidak’s multiple comparisons test).

#### 3.2.3. NEC Incidence and Histological Score

No control pups developed NEC (Figure 4A), while pups subjected to the NEC protocol and receiving only PBS as a vehicle had NEC incidence (≥grade 2) of 66.7% (10/15). More specifically, 46.7% (*n =* 7) of pups showed a grade 2 NEC, 13.3% (*n =* 2) a grade 3 NEC, and 6.7% (*n =* 1) a grade 4 NEC (Figure 4B) (Fisher’s exact test vs. control, *p* < 0.0001).

No signs of histological NEC were developed in experimental groups subjected to NEC and treated with the higher dose of hBM-MSCs (1 × 10^6^) (Fisher’s exact test, *p* = 0.0002, hBM-MSCs 1 × 10^6^ vs. NEC + PBS). On the other hand, pups treated with the lower dose of hBM-MSCs (NEC + hBM-MSCs (0.5 × 10^6^)) showed an incidence of 52.1% (12/23) (Figure 4A,B), with 26.1% (*n =* 6) of pups with grade 2, 21.7% (*n =* 5) with grade 3 and 4.7% (*n =* 1) with grade 4. Therefore, a dose-dependent effect of hBM-MSCs on NEC incidence was observed (Fisher’s exact test, *p* = 0.0008, NEC + hBM-MSCs (0.5 × 10^6^) vs. NEC + hBM-MSCs (1 × 10^6^)) (Figure 4A).

### 3.3. hBM-MSCs Reduced Apoptosis and Stimulated Tissue Regeneration in a Neonatal Mouse NEC Model

We further investigated the protective function of hBM-MSCs. Considering that hBM-MSCs could prevent apoptosis in damaged cells exposed to inflammatory insults, we evaluated the expression of the pro-apoptotic marker Caspase 3 on intestine sections [72].

Since the higher administered dose of hBM-MSCs completely abolished NEC incidence, we focused our analysis on the effects obtained by treatment with hBM-MSCs (1 × 10^6^). While we found several areas positive for the pro-apoptotic marker Caspase 3, we observed a reduced expression of Caspase 3 upon treatment with hBM-MSCs (1 × 10^6^). (Figure 5A). Accordingly, when we measured the expression level of the anti-apoptotic gene *Bcl2*, we observed a significant overexpression in the RNA extract from the intestines of mice treated with hBM-MSCs (*p* = 0.008) (Figure 5D, left panel).

In these mice, we also found a reduced expression of the pro-inflammatory cytokine gene IL1b (*p* = 0.028) (Figure 5D, right panel) in accordance with the anti-inflammatory function exerted by MSCs [72].

We next evaluated whether hBM-MSCs accelerated the regenerative process upon NEC injury. We stained intestine sections from the different experimental groups with ki67, a marker of active proliferation. The intestines of mice exposed to NEC + PBS showed a higher number of epithelial cells positive for Ki67. On the contrary, in the intestines of pups treated with hBM-MSCs, Ki67+ cells were limited in the intestinal crypts where highly proliferating intestinal stem cells reside (*p* = 0.048) (Figure 5B,C).

Moreover, we determined the expression level of ZO-1, one of the critical components of tight junctions (TJs), which is fundamental for mucosal repair [73]. Compared to control pups, pups exposed to NEC + PBS showed low expression levels of ZO-1. However, we found a robust increase in ZO-1 expression in mice exposed to NEC and treated with both hBM-MSCs concentrations, reaching levels similar to those observed in control pups (Control vs. NEC + PBS, *p* = 0.032; NEC + PBS vs. NEC + hBM-MSCs (0.5 × 10^6^), *p* = 0.017; NEC + PBS vs. NEC + hBM-MSCs (1 × 10^6^), *p* = 0.032) (Figure 5E).

## 4. Discussion

NEC still represents a devastating disease for preterm neonates. However, currently, preventive and therapeutic strategies are still scanty.

The study demonstrated that our new mouse model of NEC, based on administering full-term infants’ formula that was not hyperosmolar, can successfully induce the disease. We detected an incidence of NEC ≥grade 2 of 50%, divided according to histological score, into 23.1% grade 2 and 26.9% grade 3 NEC. As expected, the pathology development was associated with a lower survival rate of around 40% and a lower body weight in mice that underwent NEC protocol compared to the control group.

Using this novel protocol for NEC induction, we demonstrated that hBM-MSCs treatment reduced NEC incidence in a concentration-dependent manner, with the highest dose of cells preventing the development of the disease. Furthermore, no grade 2 or higher was reported in the analyzed intestines concerning the NEC + PBS group. Interestingly, although not statistically significant, we already observed a global reduction in NEC incidence with the lower concentration of MSCs (0.5 × 10^6^) compared to NEC + PBS mice, thus suggesting that the administration of hBM-MSCs was able to interfere with the pathology onset.

Recently, stem cell-based therapeutic approaches have been shown to reduce the incidence of NEC, representing a promising treatment [74]. Notwithstanding, their exact mechanism of action is still poorly understood [75,76].

Human BM-MSCs express high growth factor levels that play a key role in tissue regeneration [77]. Several works demonstrated that MSCs release growth factors to maintain damaged cells and induce a regenerative program in tissue-resident stem cells. For example, MSC-derived hepatocyte growth factor (HGF) was established to protect the infarcted heart [78].

Similarly, leukemia inhibitory factor (LIF) administration stimulates skeletal muscle regeneration upon injury, and epidermal growth factor (EGF) provides efficient wound healing [79,80].

In our model of NEC, hBM-MSCs’ administration was associated with the stimulation of tissue regeneration. Indeed, when we analyzed the expression of the proliferative marker Ki67, we found several areas of proliferation in the intestine of NEC + PBS mice, indicating that intestinal cells massively proliferate to repair the damaged tissue. In contrast, in mice treated with hBM-MSCs, we observed a reduced expression of Ki67 in the intestinal cells, with Ki67-positive cells located only at the basis of intestinal crypts, where intestinal stem cells reside and continuously proliferate to guarantee tissue cell turnover, suggesting that the injection of hBM-MSCs were able to accelerate the regenerative process.

Moreover, we also found a reduced expression of the pro-apoptotic factor Caspase 3 in NEC + hBM-MSCs mice. Intestines from mice injected with MSCs showed a regenerated tissue architecture with a barely detectable Caspase 3 signal. In contrast, intestines harvested from NEC + PBS mice were characterized not only by necrosis and dysmorphic tissue organization but also by the presence of several Caspase 3 positive areas, indicating a strong apoptosis induction in the damaged gut. Furthermore, the reduced expression of Caspase-3 was associated with a higher level of Bcl2 expression in hBM-MSC treated mice compared to controls (NEC + PBS), according to previous work showing a primary role of the Bcl2 pathway in the antiapoptotic function of MSCs [81]. We concluded that hBM-MSCs favored resident cell survival by preventing apoptosis, as previously reported in vitro and in vivo, mainly through direct paracrine mechanisms on primarily damaged cells or tissue-resident cells favoring regeneration [82,83,84,85]. hBM-MSCs may prevent excessive cell death by blocking the activation of a detrimental inflammatory program which could interfere with the regenerative process. In line with this, we also found a significant reduction in IL1b expression upon hBM-MSC administration compared to control mice (NEC + PBS).

Finally, we also demonstrated that hBM-MSCs preserved the integrity of the intestinal barrier in our model of NEC, which is fundamental for NEC resolution, considering that the intestinal epithelium offers a physical barrier that protects the host against microbial invasion [86]. Conversely, LPS administration leads to the failure of the intestinal barrier function, thus resulting in an increased mucosal permeability, which plays a key role in promoting bacterial translocation and inflammatory mediators’ release [87,88].

Epithelial tight junctions (TJs) are involved in intestinal barrier function and permeability [89]. TJs are localized on the apical membrane of epithelial cells and create a shield to the movement of the solute between cells [90]. They are dynamic structures that can be detached and reassembled in response to stimuli such as LPS and inflammatory mediators [91]. Evidence has demonstrated that TJs disruption and loss of barrier function are often associated with several diseases [86,92,93,94]. Previous studies have shown that interferon (IFN)-γ-induced damage is related to intestinal barrier disruption and the downregulation of the TJ protein ZO-1, whose decreased expression is enough to determine barrier dysfunction [94].

We hypothesized that the protective role exerted by hBM-MSCs on NEC onset and development could be related to intestinal barrier regeneration. Therefore, we analyzed ZO-1 expression in NEC mice treated with different doses of hBM-MSCs compared to vehicle and breastfed groups.

Our results showed that hBM-MSCs could act on intestinal barrier function by promoting ZO-1 expression in a concentration-dependent manner. In fact, we revealed similar ZO-1 mRNA expression levels between the highest stem cell dose and the breastfed group, suggesting that the most elevated hBM-MSCs’ dosing enabled the counteracting of the disease-induced ZO-1 downregulation.

These findings suggest that hBM-MSCs, through their paracrine action, could protect from NEC onset by promoting intestinal barrier recovery by stimulating ZO-1 expression.

Altogether, our results indicated that hBM-MSCs reduced the level of apoptosis, prevented the strong induction of IL1b, and accelerated the regenerative process in our novel experimental model of human NEC.

There are some limitations to our study. First, since NEC is a complex multifactorial disease, none of the many different animal models developed so far can fully recapitulate the underlying pathophysiology of the disease [95]. Consistently, the present model could fail to reproduce all the pathological hallmarks of the disease. In addition, the decrease in body weight observed in all the pups subjected to experimental NEC, regardless of the treatment, is likely due to the technical difficulties inherent in artificial feeding, which were extensively described by many authors [18,20,96]. Moreover, we should underline that in our model, we used a full-term infant formula, which is not hyperosmolar and utterly different in terms of nutrient composition from murine milk, as reported in Appendix A.

This could explain why, even if pups were fed every three hours with increasing daily doses for three days, they were always smaller than their breastfed counterparts.

Another limitation is that our experiments were designed to assess the changes occurring in mouse pups at the end of the NEC induction protocol. Therefore, it could not evaluate all the intestinal injuries and their recovery after different treatments.

One additional limitation is the lack of examination into the specifics of mucosal immunity and its significance in developing NEC. In particular, we did not address the role of secretory immunoglobulin A (sIgA), and given its recognized benefits in breast milk, further investigation is required to better understand its contribution to the development of NEC and its potential as a preventive approach [97]. On the other hand, we did not assess the immunological status of the intestinal milieu (in terms of immune cell subpopulations and proliferation) and how it can be affected by BM-MSCs.

Further experiments are needed to better characterize the pathways involved in the immunomodulatory activity of BM-MSCs and their mechanism of action in intestinal restitution in the context of NEC.

## 5. Conclusions

Currently, several strategies are used to prevent NEC, such as minimal enteral feeding, prebiotics, probiotics, postbiotics, and breastfeeding [7]. However, morbidity and mortality remain significantly high [98].

Our data showed that hBM-MSCs could reduce NEC incidence in a concentration-dependent manner. Thus, these findings are translationally relevant as they pave the way for new potential MSC-based preventive and therapeutic strategies to be used in NEC clinical management.

## Figures and Tables

**Figure 1 cells-12-00760-f001:**
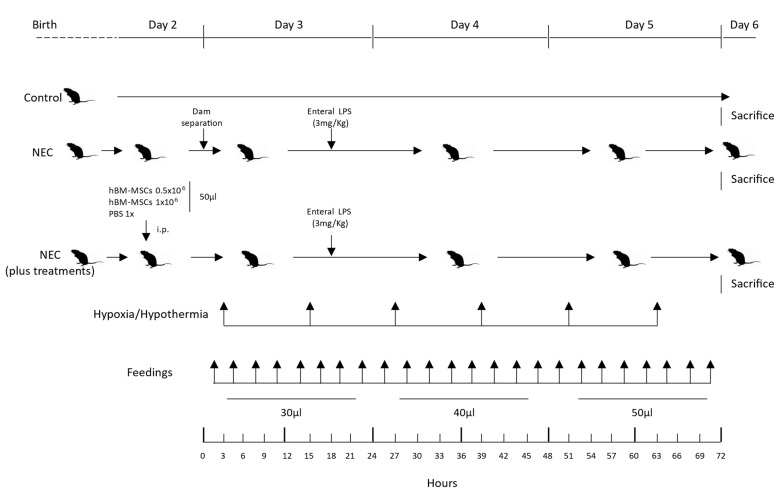
Timeline of NEC induction protocol. NEC, necrotizing enterocolitis; PBS, phosphate-buffered saline; hBM-MSCs, human bone marrow-derived mesenchymal stromal cells; LPS, lipopolysaccharide; i.p., intraperitoneal.

**Figure 2 cells-12-00760-f002:**
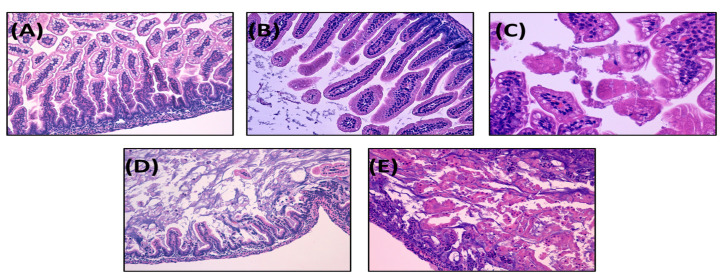
Histological grading of NEC severity in a NEC neonatal mouse model. Pups were subjected to NEC induction protocol and euthanized after 72 h or earlier in case of distress symptoms. Intestines were harvested, fixed in 10% formalin, and stained with hematoxylin–eosin. The severity of experimental NEC was classified from 0 to 4 according to a previously validated scoring system [66], as follows: (**A**) grade 0: intact villi; (**B**) grade 1: superficial epithelial cell sloughing; (**C**) grade 2: partially compromised villi structure, basal cells still present; (**D**) grade 3: complete villous necrosis, with basal cells still appreciable; (**E**) grade 4: transmural necrosis, basal membrane destroyed. Samples with histological scores of 2 or higher were considered positive for NEC. An injury score of 3 or 4 defines severe NEC. NEC: necrotizing enterocolitis. Image magnification 100×.

**Figure 3 cells-12-00760-f003:**
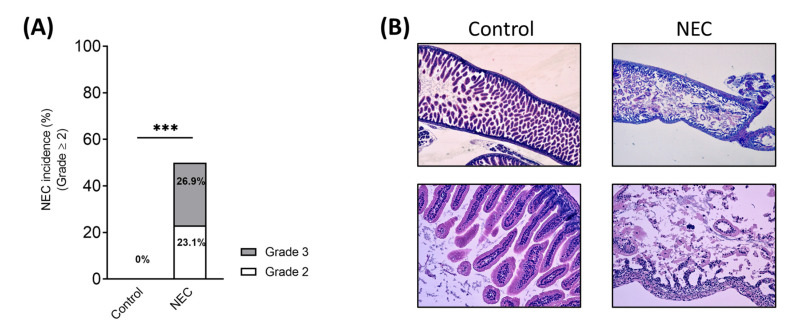
Term infant formula feedings, hypoxic–ischemic insults, and enteral LPS administration induced NEC in a neonatal mouse model. (**A**) NEC histological grading and incidence in control (*n =* 16) and NEC (*n =* 26) groups. (**B**) Representative histological sections from each group (hematoxylin/eosin staining). Image magnification: upper panels, 40×; lower panels, 100×. LPS: lipopolysaccharide; NEC: necrotizing enterocolitis. *** *p* ≤ 0.001 (Fisher’s exact test).

**Figure 4 cells-12-00760-f004:**
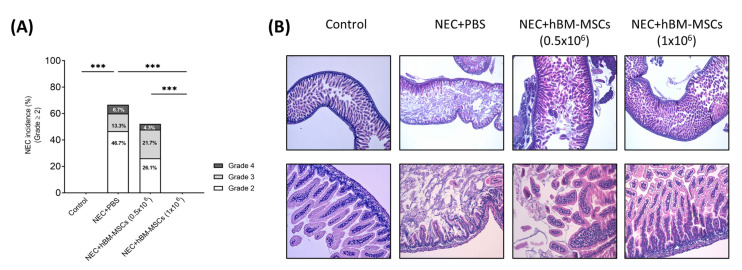
hBM-MSCs reduced NEC incidence and severity in a concentration-dependent manner in a neonatal mouse NEC model. (**A**) NEC incidence in all the experimental groups based on NEC histological grade (Control, *n =* 16; NEC + PBS, *n =* 15; NEC + hBM-MSCs (0.5 × 10^6^), *n =* 23; NEC + hBM-MSCs (1 × 10^6^), *n =* 15). (Control vs. NEC + PBS; NEC + PBS vs. NEC + hBM-MSCs (1 × 10^6^); NEC + hBM-MSCs (0.5 × 10^6^) vs. NEC + hBM-MSCs (1 × 10^6^)). (**B**) Representative histological sections from each treatment group (hematoxylin/eosin staining). Image magnification: upper panels, 40×; lower panels, 100×. hBM-MSCs: human bone marrow-derived mesenchymal stromal cells; NEC: necrotizing enterocolitis; PBS: phosphate-buffered saline. *** *p* < 0.001 (Fisher’s Exact test).

**Figure 5 cells-12-00760-f005:**
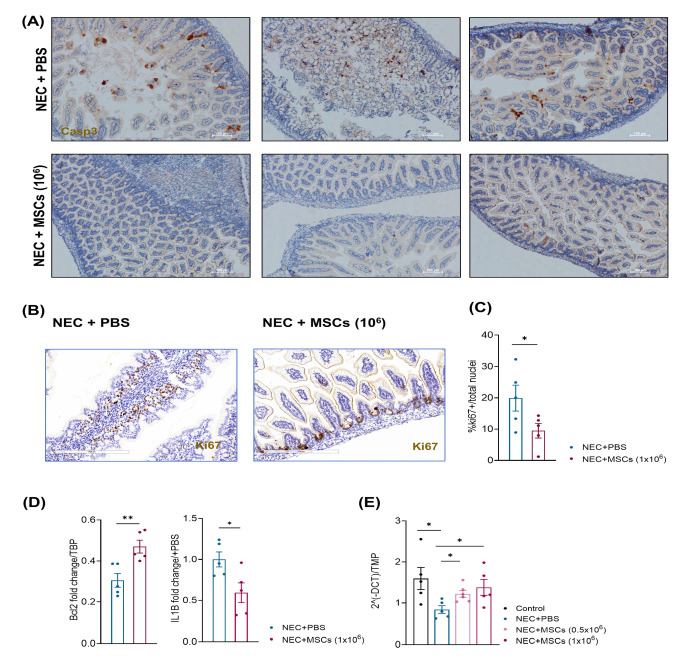
hBM-MSCs reduced apoptosis and stimulated tissue regeneration in a neonatal mouse NEC model. (**A**) Representative Caspase-3 staining on histological intestinal sections from mice belonging to the NEC + PBS and NEC + hBM-MSCs (1 × 10^6^) experimental groups. Scale bar: 100 µm. (**B**,**C**) Representative Ki67 staining and its relative quantification on histological intestinal sections from mice belonging to the NEC + PBS and NEC + hBM-MSCs (1 × 10^6^) experimental groups (*n =* 5) Scale bar: 200 µm. (**D**) Relative gene expression of the anti-apoptotic gene Bcl2 and the pro-inflammatory cytokine IL-1B. (**E**) Relative gene expression of the tight junction component ZO-1 in the four indicated experimental groups. Bcl2: B-cell lymphoma 2; hBM-MSCs: human bone marrow-derived mesenchymal stromal cells; IL-1B: Interleukin 1b; NEC: necrotizing enterocolitis; PBS: phosphate-buffered saline; ZO-1: zonula occludens-1. * *p* < 0.05; ** *p* < 0.01.

**Table 1 cells-12-00760-t001:** Primer sequences used for RT-PCR experiments.

Target	Primer Forward	Primer Reverse
Bcl-2	GAACTGGGGGAGGATTGTGG	GCATGCTGGGGCCATATAGT
IL1b	TGCCACCTTTTGACAGTGATG	TGATGTGCTGCTGCGAGATT
ZO-1	TCTTGCAAAGTATCCCTTCTGT	GAAATCGTGCTGATGTGCCA
ACTB	CGGAGTCCATCACAATGCCT	GCCATGTACGTAGCCATCCA

Bcl-2, B-cell lymphoma 2; IL1b, Interleukin 1b; ZO-1, zonula occludens-1; ACTB, actin beta.

**Table 2 cells-12-00760-t002:** Bodyweight changes (g) during experimental procedures.

PND	Control (*n =* 16)	NEC (*n =* 23)	NEC + PBS (*n =* 15)	NEC + hBM-MSCs 0.5 × 10^6^ (*n =* 23)	NEC + hBM-MSCs 1 × 10^6^ (*n =* 15)
2	1.55 (0.04)	1.54 (0.04)	1.66 (0.05)	1.66 (0.02)	1.66 (0.06)
3	1.82 (0.06)	1.81 (0.04)	2.07 (0.06) *	2.03 (0.04) *	1.92 (0.09)
4	2.13 (0.09)	1.79 (0.05) ***	1.92 (0.05)	1.93 (0.05)	1.95 (0.09)
5	2.65 (0.09)	1.77 (0.05) ***	1.79 (0.05) ***	1.81 (0.05) ***	1.86 (0.08) ***
6	3.16 (0.09)	1.69 (0.05) ***	1.75 (0.06) ***	1.85 (0.08) ***	1.74 (0.06) ***

g: grams; hBM-MSC: human bone marrow-derived mesenchymal stromal cells; n: number of pups; NEC: necrotizing enterocolitis; PBS: phosphate-buffered saline; PND: postnatal days. Variables are expressed as mean (SEM); * *p* < 0.05 vs. control; *** *p* < 0.001 vs. control. Two-way ANOVA with Sidak multiple comparisons post-hoc test.

## Data Availability

The data presented in this study are available on request from the corresponding author.

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
