# Peer review of "Human Bone Marrow-Derived Mesenchymal Stromal Cells Reduce the Severity of Experimental Necrotizing Enterocolitis in a Concentration-Dependent Manner"

_cells, 2023, doi:10.3390/cells12050760_

Round 1

Reviewer 1 Report

Thank you for the opportunity to review this article. I think it would be useful to publish this experimental study. My suggestions are:

The introduction part is very long and detailed, and this part should be shortened is accordance with the working hypothesis and method.

In selected cases, MSC treatments originating from the umbilical cord blood of another newborn can be applied as an experimental treatment in some neonatal diseases such as severe BPD, severe IVH, which are severe and do not have a curative treatment option. Why hBM-MSCs were preferred in this study should be explained and discussed in more detail.

Sincerely.

Author Response

We thank the reviewer for their comments and reduced the introduction part accordingly to the received suggestions.

In our study, we employed Bone Marrow (BM)-derived MSCs based on previous studies comparing human MSCs from different sources. While Cord Blood (CB) and BM-derived MSCs showed similar morphology, immunological privilege, immunophenotype, and proliferation capacity, the BM-MSCs most efficiently suppressed T cells and secreted the highest level of IL-10 and TGF-b (Heo, J.S. et al., Comparison of molecular profiles of human mesenchymal stem cells derived from bone marrow, umbilical cord blood, placenta, and adipose tissue. Int J Mol Med, 2016. 37(1): p. 115-25. doi.org/10.3892/ijmm.2015.2413), which plays a critical role in the resolution of tissue inflammation.

Most clinical trials used BM-derived MSCs when injected systemically (Kelly, K. and J.E.J. Rasko, Mesenchymal Stromal Cells for the Treatment of Graft Versus Host Disease. Front Immunol, 2021. 12: p. 761616. doi.org/10.3389/fimmu.2021.761616; Wiese, D.M., C.A. Wood, and L.R. Braid, From Vial to Vein: Crucial Gaps in Mesenchymal Stromal Cell Clinical Trial Reporting. Front Cell Dev Biol, 2022. 10: p. 867426. doi.org/10.3389/fcell.2022.867426; doi.org/10.1002/sctm.21-0021), showing modulation of T cell response to prevent tissue damage.

Moreover, several works demonstrated that BM-MSCs directly or indirectly favor the generation and proliferation of local progenitors, such as endothelial cells and intestine epithelial cells (Sémont, A. et al., Mesenchymal stem cells improve small intestinal integrity through regulation of endogenous epithelial cell homeostasis. Cell Death Differ, 2010. 17(6): p. 952-61. doi.org/10.1038/cdd.2009.187; Merimi, M. et al., The Therapeutic Potential of Mesenchymal Stromal Cells for Regenerative Medicine: Current Knowledge and Future Understandings. Front Cell Dev Biol, 2021. 9: p. 661532. doi.org/10.3389/fcell.2021.661532).

Based on these data, we decided to employ BM-MSCs to preserve tissue integrity in our novel model of NEC. 

Reviewer 2 Report

The manuscript entitled ‘Human bone marrow-derived mesenchymal stromal cells reduce the severity of experimental necrotizing enterocolitis in a concentration-dependent manner’, by Livia Provitera provides evidence that human BM-derived MSCs can attenuate the severity of experimental necrotizing enterocolitis (NEC) in a small animal model.

Major comments:

The study is well designed, relies on simple but adequate methods and shows decent data sets. The manuscript, however, would benefit substantially by addressing the following suggestions:

1.      In the introduction the mechanism of the underlying disease is not addressed adequately when it comes to immune cells like T cells, B cells and dendritic cells in mucosal immunity.

2.      Furthermore, the mucosal barrier upon others consists of IgA. This is not addressed, since it is known for decades that breastfeeding or alternatively oral feeding with IgA-IgG preparation can attenuate severity of NEC in humans (Wolf HM, Eibl MM. The anti-inflammatory effect of an oral immunoglobulin (IgA-IgG) preparation and its possible relevance for the prevention of necrotizing enterocolitis. Acta Paediatr Suppl. 1994;396:37-40; Eibl MM, Wolf HM, Fürnkranz H, Rosenkranz A. Prevention of necrotizing enterocolitis in low-birth-weight infants by IgA-IgG feeding. N Engl J Med. 1988 Jul 7;319(1):1-7.).

3.      The experiments relay on clinical outcome but shows no mechanism of immune cell action. Also, the immunological status of the colon is not investigated with the exemption of apoptosis/Bcl2. Which cells are affected here?

4.      The same is evident for Ki67. Which cells express Ki67? The authors should do multi-color confocal microscopy to provide more evidence.

5.      Two concentrations of MSCs 0.5/1 x 106 were applied to a mouse in group (d) and (e). This is a number where the authors should fine one or the other MSC in the colon, alternatively, microparticles of MSCs can show up. Please also address this with great care to provide more insights into the underlying mechanism.  

Minor comments:

The article is well written an does not need additional editing.  

Author Response

Major comments:

  1.  We thank the reviewer for their suggestions.

Lines 96 - 101 and 105 - 109: we provided some more information regarding mucosal immunity in the pathophysiology of NEC. As already reported, the immature intestine of preterm infants is in a hyperactive state due to abnormal bacterial colonization, ultimately leading to a robust inflammatory response and impaired intestinal perfusion, predisposing the infants to NEC. Recent research suggests that the intestinal immune system is significantly involved in this process. It consists of the intestinal epithelium, immune cells, and commensal bacteria that maintain gastrointestinal homeostasis (Hodzic, Z., A.M. Bolock, and M. Good, The Role of Mucosal Immunity in the Pathogenesis of Necrotizing Enterocolitis. Front Pediatr, 2017. 5: p. 40. doi: 10.3389/fped.2017.00040).

2. We thank the reviewer for their suggestions.

Immunoglobulin A (IgA) is the predominant antibody isotype in the mucosal immune system, widely distributed in the gastrointestinal tract, respiratory tract, vaginal tract, tears, saliva, and colostrum (Li, Y., L. Jin, and T. Chen, The Effects of Secretory IgA in the Mucosal Immune System. Biomed Res Int, 2020. 2020: p. 2032057. doi: 10.1155/2020/2032057), and they actively contribute to maintaining the balance in mucosal immunity between commensal microorganisms and mucosal surface pathogen defenses (Corthésy, B., Multi-faceted functions of secretory IgA at mucosal surfaces. Front Immunol, 2013. 4: p. 185. doi: 10.3389/fimmu.2013.00185).

However, their role in NEC still needs to be fully understood in the context of disease development and preventive strategies. Given the known beneficial properties of sIgA in breast milk, further studies are required in order to understand the role of IgA in the pathogenesis of NEC and as a viable prophylactic or treatment strategy.

3. We agree with the reviewer’s comment that immunological characterization of the intestine milieu will provide specific insights into the immunomodulatory activity of BM-MSCs in the context of NEC. However, our work aims at setting a novel model of NEC, in which we further injected BM-MSCs to investigate whether MSC-based cell therapy can prevent excessive intestinal injury as in other mouse models. In our laboratory, we are investigating the molecular mechanisms by which BM-MSCs modulate the immune environment in our newly defined model of NEC compared to cell therapy based on engineered BM-MSCs and BM-MSC-derived secretome. In particular, the immunological composition of the mouse intestine by flow cytometry will be the object of our future studies to investigate the subpopulations of T cells, monocytes, and granulocytes.

4. We thank the reviewer for their comments.

As previously reported, our aim in this work was to assess whether MSC-based therapy can reduce or prevent excessive intestinal damage due to NEC. In particular, we focused our attention on the evaluation of the regenerative potential of BM-MSCs administration in our model. As expected, based on histological observation of our samples, Ki67-positive cells are the epithelial cells that reside at the base of the intestinal crypts, as specified in lines 377-378. Interestingly, we observed an increased number of nuclei positive for Ki67 in intestines from NEC mice receiving only vehicle compared to the ones administered with BM-MSCs. This is in line with an increased stimulus for regeneration/proliferation in response to tissue injury to repair the damage. In our future works, we will consider the opportunity to perform multicolor confocal microscopy to characterize this phenomenon better.

In the manuscript, from line 495 to line 501, we addressed points 2-4 as the limitations of our study.

5. We thank the reviewer for their comments. Consistent with previous work showing that ex vivo expanded MSCs are short-lived after infusion and do not exhibit long-term engraftment, we did not detect MSCs in the intestines of treated mice (Eggenhofer, E. et al., Mesenchymal stem cells are short-lived and do not migrate beyond the lungs after intravenous infusion. Front Immunol, 2012. 3: p. 297. doi: 10.3389/fimmu.2012.00297; Eggenhofer, E., et al., The life and fate of mesenchymal stem cells. Front Immunol, 2014. 5: p. 148. doi.org/10.3389/fimmu.2014.00148).

BM-MSCs exert their anti-inflammatory and pro-regenerative functions primarily by secreting soluble factors in response to excessive damage. Only primary MSCs have been shown to engraft after injection (Abbuehl, J.P. et al., Long-Term Engraftment of Primary Bone Marrow Stromal Cells Repairs Niche Damage and Improves Hematopoietic Stem Cell Transplantation. Cell Stem Cell, 2017. 21(2): p. 241-255.e6. doi: 10.1016/j.stem.2017.07.004).

However, since MSCs represent only a small percentage of cells in the BM niche (0,001-0,1%), ex-vivo expansion is always required to achieve an adequate number of cells for pre-clinical and clinical use (Crippa, S. and M.E. Bernardo, Mesenchymal Stromal Cells: Role in the BM Niche and in the Support of Hematopoietic Stem Cell Transplantation. Hemasphere, 2018. 2(6): p. e151. doi: 10.1097/HS9.0000000000000151).

Minor comments:

Thanks